# Enhancing Protein Mutation Effect Prediction through a Retrieval-Augmented Framework

**Ruihan Guo**[*1], **Rui Wang**[*1], **Ruidong Wu**[*1], **Zhizhou Ren**[1], **Jiahan Li**[1]
**Shitong Luo**[1], **Zuofan Wu**[1], **Qiang Liu**[1,2], **Jian Peng**[1], **Jianzhu Ma**[1,3]
[1]Helixon Research, [2]The University of Texas at Austin University
[3]Institute for AI Industry Research, Tsinghua
guoruihan.sansi@gmail.com
majianzhu@tsinghua.edu.cn

## Abstract

Predicting the effects of protein mutations is crucial for analyzing protein functions and understanding genetic diseases. However, existing models struggle to effectively extract mutation-related local structure motifs from protein databases, which hinders their predictive accuracy and robustness. To tackle this problem, we design a novel retrieval-augmented framework for incorporating similar structure information in known protein structures. We create a vector database consisting of local structure motif embeddings from a pre-trained protein structure encoder, which allows for efficient retrieval of similar local structure motifs during mutation effect prediction. Our findings demonstrate that leveraging this method results in the SOTA performance across multiple protein mutation prediction datasets, and offers a scalable solution for studying mutation effects.

## 1 Introduction

Protein fitness plays significant roles in diverse applications in pharmaceutical industry [Amara, 2013], drug design [De Carvalho, 2011], biofuel production [Huang et al., 2020], and environmental bioremediation [Lu et al., 2022]. Deciphering mutation effects on protein fitness is crucial for understanding their functional dynamics and yet remains a central challenge in molecular biology. Most recent breakthroughs on computational predictions of mutation effects are driven by coevolutionary information [Riesselman et al., 2018, Luo et al., 2021, Notin et al., 2022]. The mutations on contacting residue pairs would become correlated under the evolutionary pressure to maintain protein stability and optimize functional efficiency within cellular environments. Consequently, conserved patterns within protein sequences and structures typically signify their stability and functionality. The predominant methodology to exploit such coevolutionary patterns is to perform multiple sequence alignments (MSA) [Thompson et al., 1994, 1997] and fit either statistical [Seemayer et al., 2014] or machine learning models [Rao et al., 2021]. In addition to sequence-level alignments, performing domain-level structure clustering [Orengo et al., 1997, Dong et al., 2018] is also a promising approach to extract evolutionary information from a protein family.

In this paper, we explore an alternative perspective to retrieve information for mutation effect prediction. In contrast to the global protein representation considered by MSA and domain-level structure alignments, we extract coevolutionary information in the scope of local microenvironments. We focus on the alignment of local structure motifs, *i.e.*, a central amino-acid with a few contacting neighbors. Such a local representation of coevolutionary information is specialized to the scenarios of protein engineering, where a common practice involves introducing a few point mutations to enhance a desired function [Shroff et al., 2020]. It is widely observed that the effects of point mutations are mainly given by the alteration of local biochemical microenvironments [Kim et al., 2011, Lu et al.,

2022]. Given these empirical insights, we propose to retrieve local structure fragments with similar backbone positions as auxiliary information for mutation effect prediction, while withdrawing the constraint on global sequence/structure similarity. This microscopic retrieval mechanism enables us to extract atom-level information from the whole protein universe rather than restricting to molecule-level instances within a certain protein family.

To elucidate the intricate details of local coevolutionary patterns in proteins, we employed a structure-based embedding approach, ESM-IF [Hsu et al., 2022], to encode local structure motifs into latent embeddings, assuming the metric space of such embeddings measures the similarity of motif structures. We preprocess the entire Protein Data Bank (PDB) [Berman et al., 2003] and build a database, we call Structure Motif Embedding Database (SMEDB), to support fast information retrieval by GPU-accelerated $k$-nearest neighbors (kNN) search. This retrieval procedure, we call Multiple Structure Motif Alignment (MSMA), is designed to extract coevolutionary information from protein fragments with similar local structure. We rigorously evaluate these extracted embeddings, focusing on their structural similarity and predictive accuracy for mutation effects. Remarkably, the embeddings derived from local coevolutionary patterns demonstrate superior performance compared to those obtained from traditional Multiple Sequence Alignments (MSA). Furthermore, the distribution of these embeddings was found to be complementary to those derived from MSA profiles, indicating that they capture distinct aspects of coevolutionary information, thereby enhancing our understanding of protein structure and function dynamics.

In addition, we introduce a novel model architecture, called Multi-Structure Motif Invariant Point Attention (MSM-IPA), to aggregate the retrieved coevolutionary motif information for predicting the structural fitness of proteins. This model, which effectively incorporates mutational information, has an outstanding capability to generalize across various mutations. It is trained to predict the change in binding free energy ($\Delta\Delta G$) on protein surfaces, an essential element for assessing protein-protein interactions. We extensively evaluate our model on a suite of widely-used protein stability and binding affinity benchmarks, including S669 [Pancotti et al., 2022], cDNA [Tsuboyama et al., 2023], and SKEMPI [Jankauskaitė et al., 2019] and demonstrate substantial improvement over baseline methods.

Our contributions are as follows:

- We develop Structure Motif Embedding Database (SMEDB), a comprehensive local structure alignment database encompassing all structures from the Protein Data Bank (PDB), which is organized to accelerate GPU-based kNN search.

- We propose Multiple Structure Motif Alignment (MSMA) to retrieve local coevolutionary motifs based on embeddings of protein structure encoders, which is shown to be complementary to classical sequence-level retrievals.

- Our model, Multi-Structure Motif Invariant Point Attention (MSM-IPA), pretrained on our novel database and retrieval mechanism, demonstrates superior performance in predicting $\Delta\Delta G$, surpassing other models on benchmark datasets, i.e., S669 and SKEMPI.

## 2 Preliminaries

**Definitions**  A protein consists of multiple residues, possibly from different chains. For each residue $i$, we represent the residue as its residue type $a_i \in \{1, \ldots, 20\}$ and its positions of backbone heavy atoms with $\boldsymbol{p}_{i,C}, \boldsymbol{p}_{i,C\alpha}, \boldsymbol{p}_{i,N} \in \mathbb{R}^3$. A structure motif $\mathcal{M}$ is a fragment of the whole protein structure with $N$ residues. For the raw structure motif $\mathcal{M}_{\text{raw}}$, we choose $N_{\text{raw}} = 256$ to match the pretraining settings and keep enough information about raw structure. For the retrieved structure motif (denoted as $\{\mathcal{M}_i\}_{i=1}^L$, where $L$ is the size of the retrieved motif set), we choose $N_{\text{retr}} = 16$ to ensure sufficient interactions from similar environments are captured.

Our approach uses ESM-IF as a pretrain model and CUHNSW for vector database implementation. Here we make a brief introduction to these two methods.

**ESM-IF**  ESM-IF [Hsu et al., 2022] is a model designed for protein sequence prediction using backbone structures. This approach treats inverse folding as a sequence-to-sequence problem with an autoregressive encoder-decoder architecture, allowing the model to recover native sequences from backbone atom coordinates. By incorporating a large number of sequences with predicted

structures as additional training data, ESM-IF effectively learns even when experimental structures are unavailable. The augmented data and backbone-only input make ESM-IF a suitable model to encode the nearby backbone structure of a residue.

**CUHNSW** CUHNSW [Malkov and Yashunin, 2020] is a CUDA implementation of Hierarchical NSW (HNSW), a novel approach of vector database that can do approximate K-nearest neighbor (K-NN) search utilizing navigable small world graphs with a controllable hierarchy. Unlike traditional methods, CUHNSW is fully graph-based and eliminates the need for additional search structures. It incrementally builds a multi-layer structure of proximity graphs, with elements randomly assigned to layers using an exponentially decaying probability distribution. This method enhances performance by starting the search from upper layers and leveraging scale separation, resulting in logarithmic complexity scaling. Performance evaluations show that CUHNSW outperforms previous state-of-the-art vector-only approaches and its similarity to the skip list structure allows for straightforward distributed implementation.

## 3 Methodology

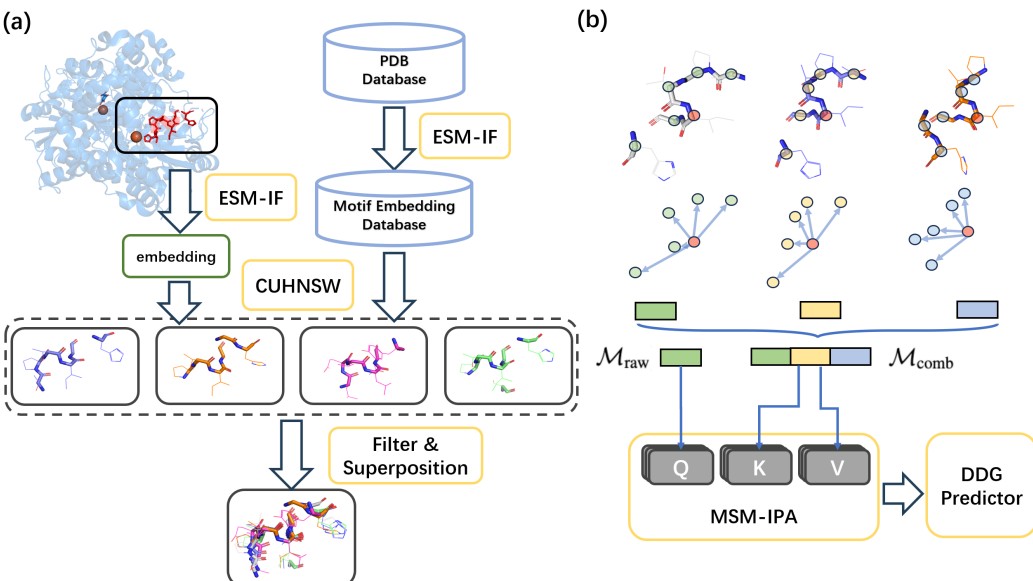

Figure 1: Overview of the retrieval augmented framework. (a) Multiple Structure Motif Alignment process. (b) A diagram illustrating MSM-Mut, a model that can predict mutation effect with information get from multiple structure motif alignment.

### 3.1 Multiple Structure Motif Alignment

We use per-residue embeddings extracted by ESM-IF model [Hsu et al., 2022] for each of the protein chains in PDB database and perform local structure search by HNSW. To obtain embeddings for ESM-IF, we utilize the encoder module named GVPTransformerEncoder in ESM-IF to generate a 512-dimensional embedding for each residue. As we exclusively use the encoder module of ESM-IF, the computational cost remains manageable, requiring approximately 3 days on 32 A100 GPUs.

The PDB dataset comprises more than 130 million residues, making it impractical to load the entire dataset into memory for querying. Traditional databases lack the capability to leverage GPU acceleration for efficiently querying the top $k$ nearest neighbors. To address this, we integrate CUHNSW as a module to obtain the $k$-nearest neighbors in the ESM-IF embedding space. The structure motifs are retrieved for their similar local interactions between residues that have similar backbone positions. The interaction between residues occurs only when they are in close proximity. Therefore, when processing the retrieved structure motifs, a large motif size is not necessary. For simplicity, in this paper, we always use $N_{retr} = 16$. With a highly efficient CUDA implementation [Yoon, 2021] the

time consumption querying the top $10^5$ neighbors of a result is about 8 seconds in 8 A100 GPUs and only 0.5 seconds for the top $10^3$ neighbors.

The advantage of using an inverse folding model as a structure encoder, compared to traditional methods that rely on retrieving from a database based on continuous structural fragments, lies in the model's inherent ability to generate embeddings tailored for predicting the surrounding environment based on a given backbone structure. The inverse folding model encodes positions within the spatial context that significantly influence the central amino acid type, rather than merely contiguous positions along the sequence.

Since we use embeddings from a structure encoder (ESM-IF in this case) instead of directly search on structures, we would like to validate the performance of the embedding retrieval method by checking the matched motif size $N_{\text{matched}}$ of the search result comparing with that of structure search. We first define the *matched* motif size as

$$N_{\text{matched}} = \sum_{i \in N_{\text{raw}}} \mathbb{1} \left[ \min_{j \in N_{\text{retr}}} \left( \text{dist}(R_{\text{raw},i}, R_{\text{motif},j}) \right) < 2.0\text{Å} \right] \tag{1}$$

The left panel of Figure 2 illustrates the distribution of alignment sizes for the retrieved structures, demonstrating that numerous similar spatial structures can be found centered on each position. The other panel of Figure 2 shows the relationship between local similarity (number of matching amino acids) and overall similarity (TM-score) with different motif size. The figure show that the similar motif may have low TM-score, indicating that our search approach can extract analogous local structures motif from structurally unrelated proteins to aid in prediction.

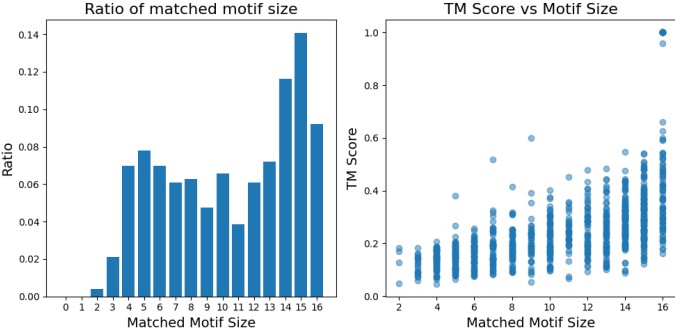

Figure 2: CUHNSW combined with embeddings from ESM-IF successfully retrieved motifs with large *matched* regions, and possibly motifs distant in sequence identity, where matched residues are defined by the indicator function in Eq. (1). Moreover, lower TM-Score is obtained for motifs with high number of locally matched residues, indicating that our search approach can extract analogous local structures motif from structurally unrelated proteins.

## 3.2 Multi Structure Motif Modelling

In this section, we introduce how we utilize the retrieved structure motifs to help predict the effects of mutations. We begin by discussing the filtering of retrieved structure motifs, followed by the superposition of retrieved data, which enables the model to learn information from diverse structures. Finally, we introduce our module, MSM-IPA, which extracts similarity information from the PDB database.

**Retrieved Structure Motif Filter** After extraction, we obtain a large volume of neighbor data (on the order of $10^3$). For efficiency reasons, we only reserve $L_{\text{filter}} = 16$ data points with most valuable information. This is done in the following manner: Firstly, retrieved structure motifs with central amino acid different from query are discarded. After that, We rank them with a scoring function and retain the top results. The scoring formula can be expressed as follows:

$$\sum_{i \in N_{\text{raw}}} \mathbb{1} \left[ \min_{j \in N_{\text{retr}}} \left( \text{dist}(R_{\text{raw},i}, R_{\text{motif},j}) \right) < 2.0\text{Å} \right] \cdot \exp(-\|p_{i,C\alpha} - p_{0,C\alpha}\|_2) \tag{2}$$

where $\mathbb{1}[\cdot]$ is the indicator function and the distance function $\text{dist}(R_1, R_2) = \|p_{1,C\alpha} - p_{2,C\alpha}\|_2$. Intuitively, we use the weighted term $\exp(-\|p_{i,C\alpha} - p_{0,C\alpha}\|_2)$ to rewrite Eq. (1) to retain as many contacts around the central amino acid as possible.

When dealing with multiple mutations, which involves several central amino acids, we ensure a balanced selection of retrieved structure motif information by using each central amino acid as a query. This approach guarantees an even distribution of the retrieved information across all central amino acids.

In the context of mutation analysis, we conduct separate selection processes for both the pre-mutation (wild type) and post-mutation states. The key distinction between these two processes is the amino acid type in the initial step—one being the wild type and the other the mutated form.

**Superposition**    The choice of superimposition method depends on the nature of the extracted information. We compare two superimposition approaches: alignment based on the central frame and alignment based on the overall structure. Experimental results indicate that the central frame-based alignment method provides better alignment quality for the retrieved structures. Consequently, we select the central frame-based alignment method for subsequent analyses.

For each residue, we can construct the frame from its backbone atom positions $\boldsymbol{p}_C, \boldsymbol{p}_{C_\alpha}, \boldsymbol{p}_N$.

$$\boldsymbol{v}_{N,C_\alpha} = (\boldsymbol{p}_N - \boldsymbol{p}_{C_\alpha})/\|\boldsymbol{p}_N - \boldsymbol{p}_{C_\alpha}\|$$
$$\boldsymbol{v}_{C,C_\alpha} = (\boldsymbol{p}_C - \boldsymbol{p}_{C_\alpha})/\|\boldsymbol{p}_C - \boldsymbol{p}_{C_\alpha}\|$$
$$\boldsymbol{R} = [\boldsymbol{v}_{N,C_\alpha}, \boldsymbol{v}_{C,C_\alpha}, \boldsymbol{v}_{N,C_\alpha} \times \boldsymbol{v}_{C,C_\alpha}]$$
$$\boldsymbol{t} = \boldsymbol{p}_{C_\alpha},$$

where $\times$ is the cross product between two vectors.

Hence with the given raw structure motif $\mathcal{M}_{\text{raw}}$ and any other structure motif $\mathcal{M}_{\text{retr}}$, we firstly take out the central residue $\mathcal{R}_{\text{raw}}^C$ and $\mathcal{R}_{\text{retr}}^C$. Then we can calculate the frame of both the residues, called $(\boldsymbol{R}_C, \boldsymbol{t}_C)$ and $(\boldsymbol{R}_{\text{retr}}, \boldsymbol{t}_{\text{retr}})$ respectively. Then we can define the alignment function as:

$$\mathcal{F}_{\text{align-atom}}(\boldsymbol{p}) = \boldsymbol{R}_C \boldsymbol{R}_{\text{retr}}^T(\boldsymbol{p} - \boldsymbol{t}_{\text{retr}}) + \boldsymbol{t}_C.$$

The alignment procedure involves applying the alignment function to each atom of $\mathcal{M}_{\text{retr}}$ to obtain the aligned motif $\mathcal{M}_{\text{retr}}^{\text{aligned}}$. For simplicity, in the following sections, we will omit the term "aligned" and assume that all motifs are aligned with the raw motif by default.

**MSM-IPA**    To retrieve information from structure motifs, we propose Multi-Structure Motif Invariant Point Attention (MSM-IPA), which is inspired by and similar to the IPA module in AlphaFold2 [Jumper et al., 2021]. The MSM-IPA module takes the raw structure motif $\mathcal{M}_{\text{raw}}$ and the processed retrieved structure motifs $\mathcal{M}_1, \dots, \mathcal{M}_L$ as input. For simplicity, we merge the raw motif with the retrieved motifs into a single motif, denoted as $\mathcal{M}_{\text{comb}} = \{r \in \mathcal{M}_i, \forall i \in \{1, \dots, L\}\} \cup \{r \in \mathcal{M}_{\text{raw}}\}$. $\mathcal{M}_{\text{comb}}$ is then used as *key* motifs in the cross attention mechanism. The size of this combined motif is represented as $N_{\text{comb}}$.

To extract information from the motif, we define two encoders, $\text{Enc}_s(\mathcal{M})$ for single node-wise representations and $\text{Enc}_z(\mathcal{M}_1, \mathcal{M}_2)$ for pair edge-wise representations. The single encoder encodes the residue types and positions of atoms into a single representation, $\boldsymbol{s}$. The pair encoder encodes information such as relative positions in the sequence, spatial relative positions, and amino acid type pairs into a pair representation, $\boldsymbol{z}$.

In MSM-IPA, we extract scalar and vector representations from $\boldsymbol{s}$. The attention weights are updated and aggregated from three sources of information: single, pair, and predicted points. This approach ensures that the model efficiently utilizes the information while maintaining invariance to overall rotation and translation. The updated single representation of the raw structure motif is obtained by concatenating the weighted sums of each data representation. The detailed implementation of this algorithm is shown in Algorithm 1.

**MSM-Mut**    The MSM-Mut model mainly consists of two parts: a series of MSM-IPA to fusion the information and a mutation effect predictor consisted of a series of MLP. For both the wild-type and mutated structures, we retrieve structure motifs independently, merge the information using MSM-IPA, and model it with IPA to obtain the single representations for both structures. These representations are then fed into the mutation effect predictor to get the final impact on the structure.

### 3.3 Model Training Details

Our model training comprises two phases. In the pretraining phase, we utilize a meticulously curated dataset from the Protein Data Bank-REDO (PDB-REDO) [Joosten et al., 2014] for our pretraining data. The dataset is split into training, validation, and test sets in a ratio of 95%:0.5%:4.5%. The pretraining process involves an initial 200,000 steps without the inclusion of retrieved structure motifs, followed by an additional 30,000 steps incorporating retrieved structure motifs. During training, a random amino acid was selected, and its 256 nearest amino acids are extracted with their amino acid types and backbone atom positions. The type of the central amino acid was masked since the model was tasked with predicting it.

In the finetuning phase, to ensure comparability of our model, we follow the settings of StableOracle and RDE. For the stability mutation effect prediction task, we use the CDNA120K [Tsuboyama et al., 2023] subset of the training set, which was deduplicated against s669 in StableOracle. For the PPI surface mutation effect prediction task, we perform 3-fold cross-validation on the SKEMPI dataset, partitioned by PDBID.

## 4 Results

In this section, we show that our model outperforms other models on a series of tasks with the retrieved MSM information. To begin with, in section 4.1.1, we demonstrate that the retrieved information provides an advantage in predicting mutation effects on protein-protein interfaces. Subsequently, in section 4.1.2, we present a case study on antibody engineering for SARS-CoV-2, illustrating our model's strong performance on real-world targets. In addition, in section 4.2.1, we conducted an experiment to show that our model outperforms others on the protein stability change dataset. To further validate our model's capability in stability prediction, in section 4.2.2, we tested it on a novel enzyme dTM dataset and achieved excellent results.

**Baseline models** The baselines referenced in this study encompass two main categories: unsupervised models, and semi-supervised/supervised models. The unsupervised models comprise PSSM (position-specific scoring matrix), ESM-1v [Meier et al., 2021], MSA Transformer [Rao et al., 2021], Tranception [Notin et al., 2022], ESM-IF [Hsu et al., 2022], ProteinMPNN [Dauparas et al., 2022], Rosetta [Park et al., 2016, Alford et al., 2017], and FoldX [Delgado et al., 2019]. The semi-supervised category includes MIF-Net [Yang et al., 2023], RDE-Net [Luo et al., 2023], and DiffAffinity [Liu et al., 2024], while supervised models encompass DDGPred [Shan et al., 2022] and the End-to-End method.

**Evaluation metrics** In this section, we employ six metrics to evaluate our model: Pearson and Spearman correlations to quantify overall trends, per-structure metrics to assess the model's ability to identify beneficial and detrimental mutations within each structure, and RMSE and MAE to measure numerical differences in predicted energy values. Among these, the most critical metric is the per-structure correlation, as practical applications often involve ranking different mutations within the same structure.

### 4.1 Mutation Effects Prediction on PPI Surface

Following the settings in RDE [Luo et al., 2023], we partitioned the dataset into three folds based on PDB IDs. Two of these folds were further divided into training and validation sets in a 95:5 ratio based on PDB IDs, while the remaining fold was used as the test set. By training in this manner, we obtained three distinct sets of model parameters, each corresponding to the performance measured on the entire dataset used as the test set.

### 4.1.1 Experimental Results on SKEMPI2.0

This experiment primarily demonstrates two key points. Firstly, we establish a simple baseline that directly utilizes the top 100 retrieved structure motifs obtained from our search to construct a profile, *i.e.*, simple statistics of the amino-acid types according to the retrieval results. This baseline, called MSM-profile, achieves significantly better performance compared to MSA, and even surpass the majority of unsupervised language models. Note that MSM-profile is built upon the embeddings

obtained from the ESM-IF model, and we observed that the per-structure Pearson and Spearman correlation coefficients extracted directly are comparable to those predicted by the ESM-IF model. This indicates that our search method effectively retains the information from ESM-IF, allowing us to explicitly extract local structure motifs corresponding to the ESM-IF distribution.

Secondly, the semi-supervised model trained with the enhanced information achieved state-of-the-art performance across almost all metrics. Among the metrics, the per-structure correlation is the most crucial, as it reflects the model's ranking capability for mutations on PPI surfaces. We found that the use of the information we extracted resulted in a significant improvement in the model's prediction correlation, underscoring the value of the retrieved information.

Table 1: Performance comparison to baseline methods on SKEMPI2.0 benchmark.

| Category | Method | Per-Structure | | Overall | | | |
| | | Pearson | Spearman | Pearson | Spearman | RMSE | MAE |
|---|---|---|---|---|---|---|---|
| Energy function | Rosetta | 0.3284 | 0.2988 | 0.3113 | 0.3468 | 1.6173 | 1.1311 |
| | FoldX | 0.3789 | 0.3693 | 0.3120 | 0.4071 | 1.9080 | 1.3089 |
| Profile | PSSM | 0.0826 | 0.0822 | 0.0159 | 0.0666 | 1.9978 | 1.3895 |
| | **MSM-profile** | 0.1905 | 0.1886 | 0.1653 | 0.2063 | 1.9423 | 1.3784 |
| Unsupervised | ESM-1v | 0.0073 | -0.0118 | 0.1921 | 0.1572 | 1.9609 | 1.3683 |
| | MSA Transf. | 0.1031 | 0.0868 | 0.1173 | 0.1313 | 1.9835 | 1.3816 |
| | Tranception | 0.1348 | 0.1236 | 0.1141 | 0.1402 | 2.0382 | 1.3883 |
| | ESM-IF | 0.2241 | 0.2019 | 0.3194 | 0.2806 | 1.8860 | 1.2857 |
| | ESM2 | 0.0100 | 0.0100 | 0.1700 | 0.1630 | 2.6580 | 2.0210 |
| | EVE | 0.1131 | 0.0898 | 0.1237 | 0.1088 | 2.2622 | 1.4178 |
| Semi-sup./ Supervised | ESM2(Sup) | 0.3330 | 0.3040 | 0.6030 | 0.5290 | 2.1500 | 1.6700 |
| | DDGPred | 0.3750 | 0.3407 | 0.6580 | 0.4687 | 1.4998 | 1.0821 |
| | End-to-End | 0.3873 | 0.3587 | 0.6373 | 0.4882 | 1.6198 | 1.1761 |
| | MIF-Net. | 0.3965 | 0.3509 | 0.6523 | 0.5134 | 1.5932 | 1.1469 |
| | RDE-Net. | 0.4448 | 0.4010 | 0.6447 | 0.5584 | 1.5799 | 1.1123 |
| | DiffAffinity. | 0.4220 | 0.3970 | 0.6690 | 0.5560 | 1.5350 | 1.0930 |
| | MSM-Mut (w/o retrieval) | 0.4325 | 0.4031 | 0.6233 | 0.4954 | 1.6076 | 1.2155 |
| | **MSM-Mut** | **0.4736** | **0.4354** | **0.6814** | **0.5786** | **1.4703** | **1.0212** |

### 4.1.2 SARS-COV-2 Antibody Optimization

In Shan et al. [2022], five single mutations are identified that enhance the neutralization effectiveness of antibodies against SARS-CoV-2. Within the three CDR regions in the heavy chain, spanning a total of 26 positions, there are 494 possible mutations. Our task is to select these five beneficial mutations from the pool. We compare our results with a subset of the baseline methods that performed well in Section 4.3. Experimental results indicate that our model excels in predicting the ranking of RH103M, while also maintaining high ranks for the other four mutations. This improvement is attributed to our model's ability to extract similar structure motifs from the database.

Table 2: Rankings of the five beneficial mutations on an anti-SARS-CoV-2 antibody.

| Category | Method | TH31W | AH53F | NH57L | RH103M | LH104F |
|---|---|---|---|---|---|---|
| Energy function | Rosetta | **10.73%** | 76.72% | 93.93% | **11.34%** | 27.94% |
| | FoldX | **13.56%** | **6.88%** | **5.67%** | **16.60%** | 66.19% |
| DL-based | RDE-Net | **5.06%** | **12.15%** | 35.47% | 50.61% | **9.51%** |
| | DiffAffinity | **7.28%** | **3.64%** | **18.82%** | 81.78% | **10.93%** |
| | MSM-Mut (w/o retrieval) | **9.51%** | **11.94%** | 36.44% | 50.61% | 23.19% |
| | MSM-Mut | **6.48%** | **10.12%** | **16.19%** | **19.23%** | 20.04% |

**Case Study: RH103M**  Given that this task focuses on antibody optimization, we incorporated a post-processing step in the retrieval process. We prioritize those retrieved data corresponding to an antibody or nanobody and if the relevant position was located in a loop region. At position H103,

out of the top 1000 candidates, only 5 structure motifs remained after filtering, with one having the central residue type as methionine (M). Experimental results indicated that including this motif significantly influenced the outcome. Analysis revealed that this structure corresponds to residue 576 of chain T in the 5KOV structure. The aligned structures are depicted in Figure 3.

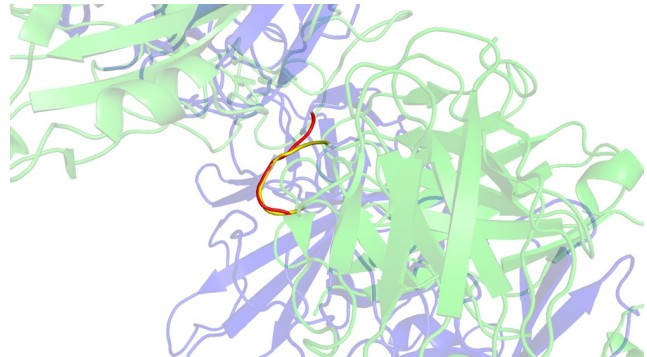

Figure 3: A diagram illustrating a set of highly similar local antibody structures obtained through Multiple Structure Motif Alignment. The figure compares the local structures of the T chain 576 from 5KOV and the H chain 103 from 7FAE.

This demonstrates that our data retrieval method can effectively assist in extracting relevant local structures for antibody design, potentially offering an interpretable approach to the antibody engineering process.

## 4.2 Stability Change Prediction

**Training MSM-Mut(or other model name) for protein engineering**    To train the model's affinity prediction module, we fine-tuned the stability prediction component using the cDNA dataset. To fully utilize the mutation data and reduce bias, ensuring a more balanced representation of mutation types, we followed the settings described in Stability Oracle . Following Diaz et al. [2023], we use Thermodynamic Permutations (TP) to perform data augmentation to balance the mutation type distribution of dataset cDNA120K [Tsuboyama et al., 2023] and train on 2M thermodynamically valid ddG measurements. TP achieves this by exploiting properties of Gibbs free energy and dataset characteristics. Specifically, the original 19 mutations at each position are augmented to cover all 380 $(19 \times 20)$ pairwise mutations.

### 4.2.1 Experimental Results on S669

In this study, we utilized a newly curated dataset (S669) [Pancotti et al., 2022] derived from the latest version of ThermoMutDB. This dataset comprises 669 protein variants not found in commonly used training sets, offering a robust basis for evaluating prediction models. Although our model does not match Stability Oracle in predicting stability changes, the integration of retrieved structure motifs enabled our model to surpass the such methods, thereby establishing a new state-of-the-art in this task.

### 4.2.2 Thermostability Optimization on Novozymes Dataset

To demonstrate our model's robust generalization capability on new data, we tested it on a novel enzyme thermostability dataset provided by Novozymes [Pultz et al., 2022]. This dataset includes experimental measurements of melting temperature of point mutations on a novel enzyme sequence that has no high-similarity match in PDB database with sequence identity higher than 30%. An AlphaFold prediction of wild-type protein structure is released with the dataset as the reference to perform structure-based methods. We adopt this predicted structure to retrieve local motifs and feed to MSM-Mut. As the results shown in Table 3b, our method significantly outperforms both classical force fields and machine-learning-based baselines. It suggests that our method can be applied to novel proteins and predicted structures.

Table 3: Performance comparison to baseline methods on S669 and Novozymes datasets.

(a) Results on S669 dataset.

| Method | Pearson | RMSE |
|---|---|---|
| ESM-1v | 0.16 | 3.05 |
| ESM-IF | 0.27 | 2.43 |
| FoldX | 0.22 | 2.30 |
| Rosetta | 0.39 | 2.70 |
| ProteinMPNN | 0.26 | 3.32 |
| Stability Oracle | 0.52 | 1.43 |
| MSM-Mut (w/o retrieval) | 0.45 | 1.73 |
| MSM-Mut | **0.54** | **1.51** |

(b) Results on Novozymes dataset.

| Method | Spearman |
|---|---|
| ESM-1v | 0.174 |
| ESM-IF | 0.255 |
| FoldX | 0.415 |
| Rosetta | 0.438 |
| ProteinMPNN | 0.231 |
| MSM-Mut (w/o retrieval) | 0.323 |
| MSM-Mut | **0.484** |

## 5  Related Work

**Mutation effect prediction**  Mutation effect prediction plays a pivotal role in in-silico protein engineering. There are three major categories of methods for mutation effect prediction–biophysical [Schymkowitz et al., 2005, Park et al., 2016, Alford et al., 2017, Steinbrecher et al., 2017], statistical [Geng et al., 2019, Zhang et al., 2020], and deep learning-based methods [Rao et al., 2021, Liu et al., 2021, Shan et al., 2022, Yang et al., 2023, Luo et al., 2023].

Different information is provided as inputs to different deep learning models. Models based on protein language models (PLMs) usually require only the primary sequence of both wild type and mutation [Meier et al., 2021, Notin et al., 2022]. MSAs also serve as inputs to mutation prediction models [Hopf et al., 2017, Riesselman et al., 2018, Rao et al., 2021, Luo et al., 2021, Frazer et al., 2021] based on the observation that co-evolutionary information correlates with the information needed. In addition to sequences, structures of protein also provide additional information. Along this line, a number of models are pretrained on protein structure data to encode the structure information [Hsu et al., 2022, Yang et al., 2023, Zhang et al., 2023]. However, MLStrA-IPA is, to our knowledge, the first model that employs multiple structures for single mutation effect prediction.

**Retrieval-based deep learning**  Retrieval-based deep learning has made its splash in language modeling recently [Guu et al., 2020, Wu et al., 2022b] that achieves higher performance-per-parameter.

Retrieval-based predictions have been long employed in protein structure prediction where people searches for evolutionarily related sequences of the sequence of interest and gather them into one MSA input to a model, which has been an important building block for models including AlphaFold [Jumper et al., 2021] and RoseTTAFold [Baek et al., 2021], while replacing MSA modules with language model demonstrates a small but noticeable drop in performance [Wu et al., 2022a, Lin et al., 2023].

Retrieval mechanism other than MSA has also been investigated previously. Retrieved Sequence Augmentation [Ma et al., 2023] augments PLMs with a collection of related sequences, either in sequence or in structure, and claims substantial improvements over MSA Transformer [Rao et al., 2021] both in performance and in prediction speed. Wang et al. [2023] use retrieval techniques to improve the performance of controllable molecule generation.

## 6  Conclusion

In this study, we addressed the challenge of predicting the effects of protein mutations by introducing a novel retrieval-augmented framework that leverages local structure motifs. By creating the Structure Motif Embedding Database (SMEDB) and using Multiple Structure Motif Alignment (MSMA), we efficiently retrieved and utilized local coevolutionary information. Our Multi-Structure Motif Invariant Point Attention (MSM-IPA) model demonstrated superior performance on benchmark datasets, proving the value of focusing on local structural environments. Our model offers a scalable and robust method for studying protein mutations, with important implications for protein engineering and genetic disease research.

# 7 Acknowledgements

This work is supported by the National Key RD Program of China No.2021YFF1201600

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

# A   Resources

The code is available at https://github.com/guoruihan/MSM-Mut

# B   Comparison of MSA-Profile and MSM-Profile in Predicting Mutation Impact

The conservation of Multiple Sequence Alignment (MSA) in sequences is relatively strong, but when dealing with surface mutation data and predicting the impact of mutations on binding affinity, sequence-based profiles tend to be weaker. Our MSM-Profile addresses this limitation by leveraging local structure motifs, which can share information regardless of whether they are intra-chain or inter-chain.

Table 4 presents the performance results of our MSM-Profile and the traditional MSA-Profile on the SKEMPI2.0 dataset. As shown, our MSM-Profile exhibits a natural advantage in this task.

Table 4: Performance comparison between MSA-Profile and MSM-Profile on the SKEMPI2.0 dataset.

| Category | Method | Pearson (P.S.) | Spearman (P.S.) | Pearson | Spearman |
|----------|--------|----------------|-----------------|---------|----------|
| Profile | MSA-Profile | 0.0826 | 0.0822 | 0.0159 | 0.0666 |
| | MSM-Profile | 0.1551 | 0.1766 | 0.1433 | 0.1739 |
| | MSM-Profile (Filtered) | **0.1905** | **0.1886** | **0.1653** | **0.2063** |

Additionally, on the s669 dataset, although the distributions of our MSM-Profile and MSA-Profile are similar, combining the two profiles results in an increase in Pearson correlation. This finding suggests that each profile provides complementary information, enhancing overall model performance when both are used together.

Table 5: Performance of MSA-Profile, MSM-Profile, and their combination on the s669 dataset.

| Method | Pearson |
|--------|---------|
| MSA-Profile | 0.17 |
| MSM-Profile | 0.19 |
| MSA-Profile + MSM-Profile | **0.23** |

# C   Ablation Study

## C.1   Interpretability and Importance of Retrieved Motifs in Mutation Prediction

In this study, we use ESM-IF embeddings to construct a database and retrieve Multi-Structure-Motifs (MSM) to support mutation effect predictions. We demonstrate that using only the top-1 neighbor provides interpretability benefits, as retrieval with ESM-IF embeddings effectively identifies information valuable for mutation effect prediction.

To assess how varying the number of retrieved neighbors impacts predictive performance, we conducted experiments on the s669 dataset. Results indicate that increasing the number of neighbors improves the model's predictive ability. This finding suggests that the top-ranked neighbors contain diverse, biologically relevant information that enhances prediction accuracy.

## C.2   Ablation Study on Retrieval and Pre-training

The table below presents an ablation study comparing the performance differences when retrieval is omitted, pre-training is omitted, or both are omitted. We observed that pre-training an IPA module is crucial. Without pre-training, the model lacks a proper initial distribution for the 20 types of amino acids at masked positions, which negatively impacts subsequent tasks.

Table 6: Performance of MSM-Mut with varying numbers of neighbors on the S669 dataset.

| Method | Pearson | RMSE |
|---|---|---|
| MSM-Mut (w/o retrieval) | 0.45 | 1.73 |
| MSM-Mut (1 neighbor) | 0.49 | 1.62 |
| MSM-Mut (2 neighbors) | 0.51 | 1.57 |
| MSM-Mut (4 neighbors) | 0.53 | 1.53 |
| MSM-Mut (8 neighbors) | 0.53 | 1.55 |
| MSM-Mut (16 neighbors) | **0.54** | **1.51** |
| MSM-Mut (32 neighbors) | 0.54 | 1.52 |
| MSM-Mut (1024 neighbors) | 0.51 | 1.63 |

Table 7: Ablation study on the impact of retrieval and pre-training on the S669 dataset.

| Method | Pearson | RMSE |
|---|---|---|
| MSM-Mut (w/o retrieval, w/o pretrain) | 0.37 | 2.82 |
| MSM-Mut (w/o pretrain, 16 neighbors) | 0.43 | 2.15 |
| MSM-Mut (w pretrain, w/o retrieval) | 0.45 | 1.73 |
| MSM-Mut (w pretrain, 16 neighbors) | **0.54** | **1.51** |

## C.3 Impact of Retrieval Dataset Size on Model Performance

To illustrate the impact of retrieval dataset size on model performance, we present results based on random selections of approximately one-tenth, one-hundredth, and one-thousandth of the database. Specifically, we randomly selected 16 motifs from the top 100, 1000, and 10,000 motifs, simulating a reduction of the database to approximately one-tenth, one-hundredth, and one-thousandth of its original size. Our observations indicate that within the existing high-quality structure database, the size of the dataset is indeed critical, as it significantly influences the availability of high-quality data.

Table 8: Impact of database size on model performance for the S669 dataset.

| Method | Pearson | RMSE |
|---|---|---|
| MSM-Mut (w/o retrieval) | 0.45 | 1.73 |
| MSM-Mut (random in top 10000) | 0.43 | 2.03 |
| MSM-Mut (random in top 1000) | 0.52 | 1.57 |
| MSM-Mut (random in top 100) | 0.53 | 1.59 |
| MSM-Mut (top 16 neighbors) | **0.54** | **1.51** |

## D Comparison of Continuous Backbone Angle Embedding (CBAE) and ESM-IF Embeddings

To provide a better comparison, we implemented a simpler encoding method called Continuous Backbone Angle Embedding (CBAE). For this approach, we defined the $\phi$ and $\psi$ angles corresponding to amino acid $i$ as $\phi_i$ and $\psi_i$. For each amino acid, we created an 18-dimensional embedding vector $[\phi_{i-4}, \psi_{i-4}, \ldots, \phi_{i+4}, \psi_{i+4}]$, where each value ranges from $[-\pi, \pi]$. This vector consists of the angles for the amino acid itself and the four consecutive amino acids before and after it in the sequence. We calculated the distance between two embeddings using the Manhattan distance, ensuring that retrieved structures have similar local sequence backbone structures.

Using this retrieval method, we tested model performance on the SKEMPI2.0 and S669 datasets, as shown in Tables 9 and 10. Results indicate that, while structures retrieved using ESM-IF embeddings tend to perform better, CBAE still maintains reasonable backbone similarity.

Our experiments showed that structures retrieved using ESM-IF embeddings tend to perform better than CBAE in capturing relevant mutation information, as ESM-IF embeddings can capture non-adjacent backbone atom positions to some extent. However, the differences between these methods

Table 9: Performance of MSM-Mut with and without CBAE retrieval on the SKEMPI2.0 dataset.

| Method | Pearson (P.S.) | Spearman (P.S.) | Pearson | Spearman | RMSE |
|---|---|---|---|---|---|
| MSM-Mut (w/o retrieval) | 0.4325 | 0.4031 | 0.6233 | 0.4954 | 1.6076 |
| MSM-Mut (CBAE retrieval) | 0.4619 | 0.4262 | 0.6524 | 0.5158 | 1.5531 |
| MSM-Mut | **0.4736** | **0.4354** | **0.6814** | **0.5786** | **1.4703** |

Table 10: Performance of MSM-Mut with and without CBAE retrieval on the S669 dataset.

| Method | Pearson | RMSE |
|---|---|---|
| MSM-Mut (w/o retrieval) | 0.45 | 1.73 |
| MSM-Mut (CBAE retrieval) | 0.52 | 1.55 |
| MSM-Mut | **0.54** | **1.51** |

are limited, as both embedding methods ensure a high degree of backbone similarity in retrieved structures.

# E   Visualization of Retrieved Structures

To further illustrate the retrieval process, we provide a figure with two subfigures detailing the alignment and visualization of highly similar local antibody structures obtained through Multiple Structure Motif Alignment.

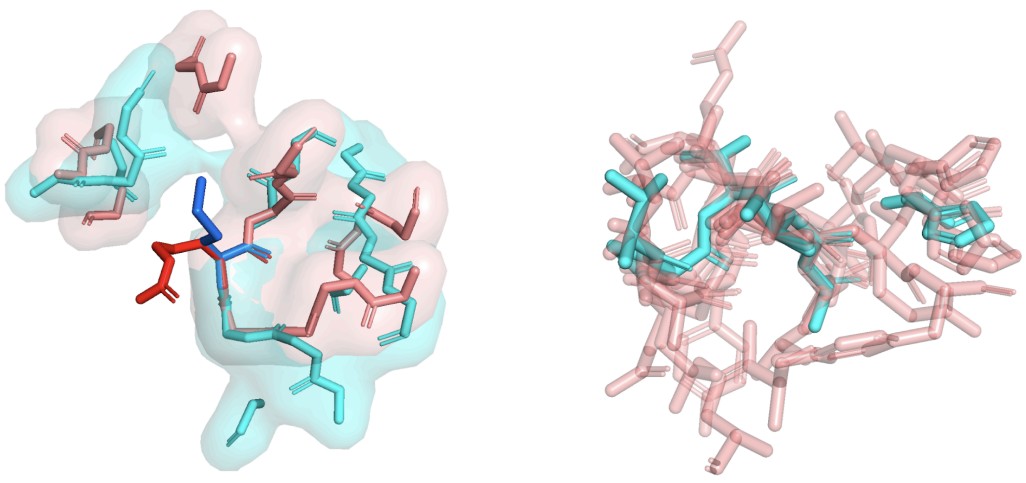

Figure 4: (a) Alignment centered on T chain 576 from 5KOV and H chain 103 from 7FAE, illustrating a high degree of similarity in their local structure motifs. (b) Visualization of the deduplicated retrieved local structure motifs, highlighting that similar structures are not limited to continuous chain segments; backbone atoms that are spatially close, even if not sequentially adjacent, also exhibit structural similarity.

# F   Details of Algorithms

**Algorithm 1** MSM-IPA Information Fusion Algorithm

---

**Require:** Raw motif $\mathcal{M}_{\text{raw}}$, Retrieved motifs $\{\mathcal{M}_i\}_{i=1}^{L}$
**Ensure:** Merged motif representation $\tilde{\mathbf{s}}_i$
 1: Combine raw and retrieved motifs:

$$\mathcal{M}_{\text{comb}} = \{\text{residue} \in \mathcal{M}_i, \forall i \in \{1, \dots, L\}\} \cup \{\text{residue} \in \mathcal{M}_{\text{raw}}\}$$

 2: Encode single representations:

$$\mathbf{s}_{\text{raw}} = \text{Enc}_s(\mathcal{M}_{\text{raw}})$$
$$\mathbf{s}_{\text{comb}} = \text{Enc}_s(\mathcal{M}_{\text{comb}})$$

 3: Encode pair representation:

$$\mathbf{z} = \text{Enc}_z(\mathcal{M}_{\text{raw}}, \mathcal{M}_{\text{comb}})$$

 4: Linear transformations:

$$q_i^h, \vec{q}_i^h = \text{LinearNoBias}(\mathbf{s}_{\text{raw}})$$
$$k_i^h, v_i^h, \vec{k}_i^h, \vec{v}_i^h = \text{LinearNoBias}(\mathbf{s}_{\text{comb}})$$
$$b_{ij}^h = \text{LinearNoBias}(\mathbf{z}_{ij})$$

 5: Compute attention weights:

$$a_{ij}^h = \text{softmax}_k\left( w_L \left( \frac{1}{\sqrt{c}} \mathbf{q}_i^{h\top} \mathbf{k}_j^h + b_{ij}^h - \frac{\gamma^h w_C}{2} \sum_p \left\| T_i \circ \overrightarrow{\mathbf{q}}_i^{hp} - T_j \circ \overrightarrow{\mathbf{k}}_j^{hp} \right\|^2 \right) \right)$$

 6: Compute outputs:

$$\tilde{\mathbf{o}}_i^h = \sum_j a_{ij}^h \mathbf{z}_{ij}$$
$$\mathbf{o}_i^h = \sum_j a_{ij}^h \mathbf{v}_j^h$$
$$\overrightarrow{\mathbf{o}}_i^{hp} = T_i^{-1} \circ \sum_j a_{ij}^h \left( T_j \circ \overrightarrow{\mathbf{v}}_j^{hp} \right)$$

 7: Compute final representation:

$$\mathbf{s}_{\text{updated-raw}} = \text{Linear}\left( \text{concat}_{h,p}\left( \tilde{\mathbf{o}}_i^h, \mathbf{o}_i^h, \overrightarrow{\mathbf{o}}_i^{hp}, \left\| \overrightarrow{\mathbf{o}}_i^{hp} \right\| \right) \right)$$

---

