# OpenReview forum: "Enhancing Protein Mutation Effect Prediction through a Retrieval-Augmented Framework"
_NeurIPS.cc/2024/Conference — NeurIPS 2024 poster_

### Official Review · Reviewer_95k8 · 2024-07-09

**Soundness:** 3
**Presentation:** 2
**Contribution:** 3
**Rating:** 6
**Confidence:** 3

**Summary:**

This paper presents a novel approach for mutation effect prediction using a retrieval-augmented framework. The main contributions involved:
* Structure Motif Embedding Database (SMEDB): a vector database storing ESM-IF local structure motif embeddings from experimentally determined protein structures in Protein Data Bank.
* Multiple Structure Motif Alignment (MSMA): an approach to efficiently retrieve and filter local coevolutionary motifs from SMEDB.
* MSM-IPA: a new model architecture leveraging retrieved coevolutionary motif information to predict the effect of mutations.
* The approach was validated on multiple benchmarks for mutation effect prediction and a case study on SARS-COV-2 Antibody Optimization.

**Strengths:**

* **Originality**: The use of a retrieval mechanism with vector databases of local structural motifs is a novel strategy for mutation effect prediction, departing from previous methods that rely on MSAs or domain-level structure clustering.
* **Quality**:  The overall quality of the paper is high. The methodology is clear, leveraging the PDB to create a comprehensive database of local structural motifs and presenting a reasonable retrieval approach, filtering and model architecture. The evaluation is thorough, involving benchmarks on protein stability, binding affinity, and a case study on antibody engineering.
* **Clarity**: The paper is well-structured, with a clear explanation of the motivation behind focusing on local structural motifs and providing a detailed discussion of the experiments. Despite missing details and ablations on retrieval hyper-parameter choices, the methods are reasonably well described.
* **Significance**: The paper makes significant methodological contributions to mutation effect prediction, with potential for other researchers to build upon.

**Weaknesses:**

*  **Scalability**: The authors claim their approach is scalable but the paper lacks experimental evidence on the impact of increasing the number of encoded residues in the retrieval database. There's no analysis in increasing/decreasing the database size or including predicted structures from AlphaFoldDB. Predicted structures are used for encoding input proteins in the Novozymes benchmark but not for retrieval. If database size is a significant bottleneck for retrieval, this should be discussed.
*  **Choice of Embedding Method**: The reliance on ESM-IF for embedding structural motifs is not well justified, and alternative methods are not explored. Discussing how to overcome performance limitations imposed by ESM-IF would strengthen the paper.
*  **Exploration of Retrieval Hyper-parameters**: The paper lacks details on hyper-parameter choices for the retrieval mechanism. The retrieved motif length (Nretr=16) and the number of retrieved motifs (Lfilter=16) could significantly impact performance but their effect is not studied. The discussions comparing the proposed approach against MSAs should also consider MSA depth vs the number of retrieved local motifs.
*  **Applicability to Other Mutation Effect Benchmarks**: I found it surprising that the methodology is not tested on ClinVar or ProteinGym which are common mutation effect prediction benchmarks. It should be discussed if the lack of a training set for these datasets is a limitation.

 **Minor**:
*  MSM-Mut Description: The paper describes the classification head as a series of MLPs but does not specify the number of layers. Including this information would help reproducibility.
*  Training and Fine-tuning Details: Limited information on hyper-parameters and procedures for pre-training and fine-tuning. Including specifics like learning rate, batch size, and training time would help other researchers replicate the work.
*  Extending the Figure 1 legend with more details on the methodology would help understanding.

**Questions:**

* Why were predicted structures not considered to extend the database? Could the authors extend their discussion on scalability and/or include analyses on the effect of increasing/decreasing the size of the underlying dataset used for retrieval?
* What was the maximum sequence length used during the encoding of the database and during mutation effect prediction? How did you handle proteins longer than this maximum length?
* Did you consider alternative underlying residue embedding methods such as ProteinMPNN?
* In the Introduction, you claim that the distribution of local motif embeddings provides complementary information to MSA profiles. Could you clarify the basis for this claim?
* Why was the method not benchmarked on ClinVar and ProteinGym datasets? Are there specific challenges or limitations that prevented these evaluations?

**Limitations:**

The paper discusses some limitations imposed by third-party modules such as ESM-IF and CUHNSW. However, some areas could benefit from further discussion or stating the limitations on:
* Scalability: Discuss how scalable the method is and any potential challenges or limitations on retrieval when increasing the database size.
* Benchmarking on Clinvar/ProteinGym: providing reasons for not benchmarking on those datasets and discussing any potential challenges or limitations would give a more comprehensive view of the method's applicability.

---

> ### Author Response · Authors · 2024-08-07
> **Rebuttal Information**
>
> **Question 1:MSM-Mut Description: The paper describes the classification head as a series of MLPs but does not specify the number of layers. Including this information would help reproducibility.**
>
> To predict the mutation effect, we first pass the information before and after the mutation, along with the corresponding retrieved motifs, through an MSM-IPA to obtain embeddings for the mutation site. Then, we compute the difference between the features before and after the mutation. To ensure the symmetry of the mutation (i.e., the ΔΔG of type 1 to type 2 should be the opposite of type 2 to type 1), we concatenate $feat_{raw} - feat_{mutate}$​ and $feat_{mutate} - feat_{raw}$ ​, and then pass them through a four-layer MLP to predict the mutation result.
>
> **Question 2: Training and Fine-tuning Details: Limited information on hyper-parameters and procedures for pre-training and fine-tuning. Including specifics like learning rate, batch size, and training time would help other researchers replicate the work.**
>
> During the pre-training on the PDB database, we used the PDB database up to December 31, 2021. In this process, we used the AdamW optimizer with a learning rate of 3e-4, beta1=0.9, beta2=0.999, and a batch size of 128. The pre-training process was conducted on 8 A100 GPUs over five days (300k steps).
>
> For the cDNA fine-tuning, we followed the same optimizer and parameters. The fine-tuning process took approximately two hours (4000 steps) on 8 A100 GPUs to achieve the best performance. Training for a longer period led to overfitting on the cDNA dataset, which negatively affected the model's generalizability.
>
> **Question 3: Extending the Figure 1 legend with more details on the methodology would help understanding.**
>
> Subgraph (a) illustrates the process of Multiple Structure Motif Alignment. First, we use ESM-IF to obtain embeddings for all amino acids in the PDB database. The size of this library is approximately 500 GB, making it impractical to use standard retrieval methods. To improve retrieval efficiency, we index this data using CUHNSW, constructing a Motif Embedding Database. For each query, we similarly use ESM-IF to obtain embeddings for the query structure, then quickly retrieve the approximate top-k nearest neighbors using HNSW based on the corresponding embeddings. After filtering and superposition, we obtain structure motifs suitable for downstream tasks. For efficiency, we have pre-searched the retrieved motifs from the entire PDB, which also facilitates future tasks.
>
> Subgraph (b) illustrates how MSM-IPA uses retrieved structure motifs to enhance the prediction of mutation effects. In the upper part of this figure, the retrieved structure motifs are aligned with the central amino acids of the query structure motif through preprocessing. These features are concatenated and input into MSM-IPA for cross-attention, thereby enhancing the model's ability to predict mutation effects.
>
> **Question 4: Why were predicted structures not considered to extend the database? Could the authors extend their discussion on scalability and/or include analyses on the effect of increasing/decreasing the size of the underlying dataset used for retrieval?**
>
> There are two considerations for not including predicted structures: quality and efficiency. In terms of quality, we assume that the local structure motifs in the RCSB PDB database are already sufficiently dense, and additional local motifs from predicted structures are also learned from the existing database. In terms of efficiency, the embeddings we currently use with ESM-IF have 512 dimensions, meaning the computational load for retrieval is quite large. Therefore, it is not feasible to increase the dataset from the existing 200k structures to 60M. However, this approach can be attempted in the future by modifying the embedding method and reducing the dimensionality. If the data balance can be managed well, theoretically, better results could be achieved.
>
> Here, we present results using a random selection of one-tenth, one-hundredth, and one-thousandth of the database. By randomly selecting 16 motifs from the top 100, 1000, and 10000, we simulate reducing the database to approximately one-tenth, one-hundredth, and one-thousandth of its original size. We observe that in the existing high-quality structure database, the size of the database is quite important as it determines the quantity of high-quality data.
>
> Task:S669
>
>   | Method               | Pearson | RMSE|
> |----------------------|---------|----------|
> | MSM-Mut (w/o retrieval) | 0.45  | 1.73   |
> | MSM-Mut (random in top 10000) | 0.43  | 2.03   |
> | MSM-Mut (random in top 1000) | 0.52  | 1.57   |
> | MSM-Mut  (random in top 100) | 0.53  | 1.59   |
> | MSM-Mut (top 16 neighbor)| 0.54  | 1.51   |

---

> ### Author Response · Authors · 2024-08-07
> **More Rebuttal Information**
>
> **Question 5: Did you consider alternative underlying residue embedding methods such as ProteinMPNN?**
>
> ESM-IF can potentially be replaced by other methods based on C-alpha point cloud matching or ProteinMPNN. In our future development, we will attempt to train an independent local structure motif encoder to reduce the embedding dimension centered on each amino acid from 512 dimensions to 32 or even lower. This reduction will enable us to incorporate a larger amount of data, including predicted data.
>
> **Question 6: In the Introduction, you claim that the distribution of local motif embeddings provides complementary information to MSA profiles. Could you clarify the basis for this claim?**
>
> The conservation of MSA in sequences is relatively strong, but when dealing with surface mutation data and predicting the impact of mutations on binding affinity, the profile from sequences tends to be weaker. Our MSM-Profile does not have this issue because, whether it is intra-chain or inter-chain, the local structure motifs are similar and can share information. Below are the performance results of our method and the MSA profile on the SKEMPI2.0 dataset. It can be seen that our profile has a natural advantage in this task.
>
> Task: SKEMPI 2.0
>
> | Category       | Method               | Pearson (P.S.) | Spearman(P.S.) | Pearson | Spearman |
> |----------------|----------------------|---------|----------|---------|----------|
> | Profile        |MSA-Profile                 | 0.0826  | 0.0822   | 0.0159  | 0.0666   |
> |                | MSM-Profile          | 0.1551  | 0.1766   | 0.1433  | 0.1739   |
> |                | MSM-Profile(Filtered)          | 0.1905  | 0.1886   | 0.1653  | 0.2063   |
>
> Additionally, we found that on the s669 dataset, although the distributions of our MSM-Profile and MSA-Profile are similar, simply adding the two profiles results in a significant increase in Pearson correlation. This indicates that there is a portion of the information in both profiles that is independent. Therefore, we can enhance current tasks that use MSA by incorporating MSM-Profile, which may improve the model's performance.
>
> Task: S669
>
>   | Method               | Pearson |
> |----------------------|---------|
> | MSA-Profile | 0.17  |
> | MSM-Profile | 0.19  |
> | MSA-Profile + MSM-Profile | 0.23  |
>
> **Question 7: Why was the method not benchmarked on ClinVar and ProteinGym datasets? Are there specific challenges or limitations that prevented these evaluations?**
>
> In this paper, we followed the setting of the Stability Oracle for benchmarking models predicting mutation effects on stability. Within the Stability Oracle test sets, we selected the latest and most challenging dataset, s669, as our benchmark dataset.
>
> During stability fine-tuning, a significant portion of our training set, the cDNA dataset, overlaps with the ProteinGym test set. Removing these overlapping data points may weaken our benchmark results on ProteinGym; hence, we did not conduct tests on this dataset. To demonstrate our model's capability, we show significant improvements on the s669 dataset, which has no overlap with the cDNA dataset. This indicates that our model performs well.
>
> The data on ClinVar is more focused on binary classification, determining whether a mutation is pathogenic. Since both our pre-training and fine-tuning datasets do not match the ClinVar setting, it is challenging to compare our model's performance on the ClinVar dataset. In the future, we will attempt to create a new dataset based on our current model. However, our method does not consider sequence co-evolution information (MSA), so a more valuable comparison would be between MSA Profile and our MSM-Profile. We plan to conduct this experiment in future work.

---

> > ### Comment · Reviewer_95k8 · 2024-08-12
> >
> > Thank you for your comprehensive rebuttal. The revisions have addressed various concerns and enhanced the manuscript quality with the additional content. I encourage the authors to add the content of this discussion to the main manuscript or Appendix.

---

> > > ### Author Response · Authors · 2024-08-12
> > >
> > > Thank you very much for your positive feedback and for recognizing the additional experiments and detailed analysis we provided in the rebuttal. We are grateful for your thoughtful suggestions and are pleased to hear that our revisions have addressed your concerns and improved the overall quality of the paper.
> > >
> > > We appreciate your recommendation to include the content of this discussion in the main manuscript or Appendix. In response, we will thoughtfully incorporate the key points and additional details from the rebuttal into the Appendix of the manuscript. This will ensure that the insights we discussed are thoroughly documented and accessible to readers, thereby further enhancing the contribution of our work.
> > >
> > > Once again, we sincerely thank you for your valuable suggestions and guidance throughout this process.

---

### Official Review · Reviewer_pY2f · 2024-07-12

**Soundness:** 3
**Presentation:** 2
**Contribution:** 4
**Rating:** 6
**Confidence:** 3

**Summary:**

The paper presents a novel retrieval-augmented framework for enhancing the prediction of protein mutation effects, which is essential for analyzing protein functions and understanding genetic diseases. The authors design a system that incorporates similar structure information from known protein structures into the mutation effect prediction process. Central to this framework is the creation of a vector database, the Structure Motif Embedding Database (SMEDB), which stores embeddings of local structure motifs derived from a pre-trained protein structure encoder. This allows for efficient retrieval of similar local structure motifs during the prediction of mutation effects.

**Strengths:**

1.	The authors propose a structure embedding database to retrieve similar protein fragments with similar local structure.

2.	This paper develops a novel architecture, MSM-IPA, to predict the structural fitness, which shows superior performance in downstream tasks.

3.	Extensive experimental results, including predictions of protein-protein interface mutation effects, case studies in antibody engineering, and applications on protein stability change datasets, validates the effectiveness of this method.

**Weaknesses:**

1. In the field of bioinformatics, it is crucial not only to make accurate predictions but also to understand the reasons behind certain predictions. Enhancing models with interpretability features can help explain the importance of retrieved motifs for each prediction, which could be valuable.

2. A more thorough comparative analysis with existing methods, especially those that also focus on local structure motifs, could be beneficial for this paper. Expanding the discussion on how the proposed framework differs from and improves upon these methods will strengthen the novelty claims of the paper.

**Questions:**

1. Regarding the Retrieved Structure Motif Filter, could you explain why 16 motifs were chosen and whether this number is sensitive to performance variations?

2. Can the authors elaborate on how the performance and limitations of the ESM-IF and CUHNSW modules impact the overall framework? How were these modules selected, and were alternative options considered?

3. How does the model handle proteins with low sequence similarity or sparse structural motifs?

4. In Equation 2, why is the distance calculated with the 0th alpha carbon atom? Does this represent the alpha carbon atom of the central amino acid?

5. What impact does pre-training have on the predictive ability of the model? How would this method perform without pre-training?

**Limitations:**

The paper acknowledges certain limitations, but it could expand on these by discussing potential weaknesses in greater detail and suggesting more specific directions for future research. This might include exploring different protein structure encoders, expanding the database, or adapting the method for different types of mutations.

---

> ### Author Response · Authors · 2024-08-07
> **Rebuttal Information**
>
> **Question 1: In the field of bioinformatics, it is crucial not only to make accurate predictions but also to understand the reasons behind certain predictions. Enhancing models with interpretability features can help explain the importance of retrieved motifs for each prediction, which could be valuable.**
>
> Thank you for your valuable suggestion. In our paper, we primarily used ESM-IF embedding to create a database and then searched for Multi-Structure-Motifs (MSM) to help predict mutation effects.
>
> From the perspective of interpretability, we have demonstrated that using only the top-1 neighbor is beneficial, indicating that retrieving with ESM-IF embeddings can indeed find information useful for predicting mutation effects. We also trained models with different numbers of neighbors and tested them on the s669 dataset. We found that as the number of neighbors increased, the model's predictive ability improved. This suggests that the top-ranked neighbors contain diverse information that can help the model make better predictions. Based on this conclusion, we can infer that the retrieved structure motifs are biologically relevant to mutations and can serve as an effective data source.
>
> Task: S669
>
>   | Method               | Pearson | RMSE|
> |----------------------|---------|----------|
> | MSM-Mut (w/o retrieval) | 0.45  | 1.73   |
> | MSM-Mut (1 neighbor) | 0.49  | 1.62   |
> | MSM-Mut (2 neighbor) | 0.51  | 1.57   |
> | MSM-Mut (4 neighbor) | 0.53  | 1.53   |
> | MSM-Mut (8 neighbor) | 0.53  | 1.55   |
> | MSM-Mut (16 neighbor)| 0.54  | 1.51   |
> | MSM-Mut (32 neighbor)| 0.54  | 1.52   |
> | MSM-Mut (1024 neighbor)| 0.51  | 1.63   |
>
> **Question 2: A more thorough comparative analysis with existing methods, especially those that also focus on local structure motifs, could be beneficial for this paper. Expanding the discussion on how the proposed framework differs from and improves upon these methods will strengthen the novelty claims of the paper.**
>
> Of traditional methods, mTM-Align[1] introduced the concept of Multiple Structure Alignment (MStrA). As the name suggests, this method excels in traditional template searches as TM-Align. mTM-align is an extension of the pairwise structure alignment program TM-align. However, this method cannot handle local structure information.
>
> The most recent work on extracting local structure motifs is MicroMiner[2]. Searching similar local 3D micro-environments in protein structure databases with MicroMiner]. MicroMiner converts protein sequences into multiple k-mers and uses a method similar to MMseqs2 for initial filtering of k-mers from both the original and mutated sequences in the database. Finally, candidates are superimposed and structurally filtered.
>
> In comparison, our method has two main advantages: First, MicroMiner heavily relies on sequence similarity, which can be too strict. Our method can identify local structure motifs with similar backbone structures but different sequences, greatly enriching the candidates. Second, MicroMiner can only search continuous sequence segments and requires separate searches for complexes. Our method can directly search surface motifs in the database and use intra-chain motifs to enhance inter-chain mutation prediction.
>
> **Question 3: Regarding the Retrieved Structure Motif Filter, could you explain why 16 motifs were chosen and whether this number is sensitive to performance variations?**
>
> As shown in the table below, we chose 16 motifs because further increasing the number did not significantly improve performance and might introduce more noise, leading to worse predictions.
>
> Task: S669
>
>   | Method               | Pearson | RMSE|
> |----------------------|---------|----------|
> | MSM-Mut (w/o retrieval) | 0.45  | 1.73   |
> | MSM-Mut (1 neighbor) | 0.49  | 1.62   |
> | MSM-Mut (2 neighbor) | 0.51  | 1.57   |
> | MSM-Mut (4 neighbor) | 0.53  | 1.53   |
> | MSM-Mut (8 neighbor) | 0.53  | 1.55   |
> | MSM-Mut (16 neighbor)| 0.54  | 1.51   |
> | MSM-Mut (32 neighbor)| 0.54  | 1.52   |
> | MSM-Mut (1024 neighbor)| 0.51  | 1.63   |

---

> ### Author Response · Authors · 2024-08-07
> **More Rebuttal Information**
>
> **Question 4: Can the authors elaborate on how the performance and limitations of the ESM-IF and CUHNSW modules impact the overall framework? How were these modules selected, and were alternative options considered?**
>
> **ESM-IF:**
>
> ESM-IF uses the GVP-Transformer to embed all backbone atoms, and this embedding is ultimately used for amino acid type prediction, which is why we chose this approach. We observed that structures retrieved using ESM-IF embeddings mostly resemble the backbone frames of amino acids that are close in the sequence. Please refer to the  figure Rebuttal Fig. 2.
>
> One issue with ESM-IF is that, although we observed that the neighbors retrieved by ESM-IF have high local backbone similarity, directly using the distance between embeddings for retrieval is a method not defined during training. Therefore, in the future, we plan to train an independent local structure motif encoder using contrastive learning to give this distance metric a clearer meaning.
> ESM-IF can potentially be replaced by other methods based on C-alpha point cloud matching or ProteinMPNN. In our future development, we will attempt to train an independent local structure motif encoder to reduce the embedding dimension centered on each amino acid from 512 dimensions to 32 or even lower. This reduction will enable us to incorporate a larger amount of data, including predicted data.
>
> **CUHNSW:**
>
> CUHNSW is an implementation of the classic HNSW (Hierarchical Navigable Small World) algorithm on GPUs, designed to perform approximate nearest neighbor search through a pre-built graph. This approach leverages GPU acceleration to enhance the performance of the search process. CUHNSW can be replaced by other scalable and parallelizable approximate nearest neighbor algorithms, such as LSH (Locality-Sensitive Hashing), KD-trees, or similar methods, depending on the specific task requirements (such as embedding dimensions), database size, and available resources.
>
> **Question 5: How does the model handle proteins with low sequence similarity or sparse structural motifs?**
>
> In the worst-case scenario, the model may converge to a state with no neighbors, relying entirely on information obtained from pretraining without any retrieval augmentation. We have observed that such data accounts for a relatively small proportion in real PDB databases. However, to enhance the model's capabilities, we will incorporate high-quality predicted structures (suck as AFDB) into the database. To reduce the overall database size, we will perform retrieval for each newly added motif and only add those motifs that do not have very similar neighbors in the database.
>
> **Question 6: In Equation 2, why is the distance calculated with the 0th alpha carbon atom? Does this represent the alpha carbon atom of the central amino acid?**
>
> The 0th amino acid represents the alpha carbon atom of the central amino acid. In Equation 2, our formula consists of two parts. The first part $1\left[ \min_{j \in N_{\text{retr}}}(\text{dist}(R_{\text{raw}, i}, R_{\text{motif}, j})) < 2.0\text{A}\right]$ attempts to find the closest match for each amino acid in the query motif within the retrieved motif. The second part $\exp(-\|p_{i, C\alpha} - p_{0, C\alpha}\|_2)$ assigns a weight to each amino acid, with the weight decreasing as the distance increases.
>
>
>
> **Question 7: What impact does pre-training have on the predictive ability of the model? How would this method perform without pre-training?**
>
> The table below presents an ablation study comparing the performance differences when retrieval is omitted, pre-training is omitted, or both are omitted. We observed that pre-training an IPA on PDB is crucial. Without pre-training, the model lacks a proper initial distribution for the 20 types of amino acids at masked positions, which negatively impacts subsequent tasks.
>
> Task: S669
>
>   | Method               | Pearson | RMSE|
> |----------------------|---------|----------|
> | MSM-Mut (w/o retrieval, w/o pretrain) | 0.37  | 2.82   |
> | MSM-Mut (w/o pretrain, 16 neighbor) | 0.43  | 2.15   |
> | MSM-Mut (w/o retrieval) | 0.45  | 1.73   |
> | MSM-Mut (16 neighbor)| 0.54  | 1.51   |
>
> **Reference**
>
> [1] Runze Dong, Zhenling Peng, Yang Zhang, Jianyi Yang, mTM-align: an algorithm for fast and accurate multiple protein structure alignment, Bioinformatics, Volume 34, Issue 10, May 2018, Pages 1719–1725
>
> [2] Sieg J, Rarey M. Searching similar local 3D micro-environments in protein structure databases with MicroMiner[J]. Briefings in Bioinformatics, 2023, 24(6): bbad357.

---

> ### Author Response · Authors · 2024-08-13
>
> During this period, we have conducted additional experiments specifically addressing some of the weaknesses you mentioned. We hope these efforts help resolve the concerns raised. We look forward to hearing your feedback.
>
> **1. Exploring different protein structure encoders**
>
> To provide a better comparison, we tried a simpler encoding method: we defined the $\phi$ and $\psi$ angles corresponding to the amino acid i as $\phi_i$ and $\psi_i$. For an amino acid, we defined its embedding as [$\phi_{i-4}, \psi_{i-4}, ..., \phi_i, \psi_i, ..., \phi_{i+4}, \psi_{i+4}$], an 18-dimensional vector where each value ranges from [-pi, pi]. This vector consists of the angles corresponding to this amino acid and the four consecutive amino acids before and after it in the sequence. We defined the distance between two embeddings as their Manhattan distance. This method is referred to as the Continuous Backbone Angle Embedding (CBAE). This embedding method ensures that the retrieved structures have similar local sequence backbone structures. Using such retrieving methods, we tested the performance on SKEMPI and S669 datasets as follows:
>
> Task: SKEMPI 2.0
>
>   | Method               | Pearson (P.S.) | Spearman(P.S.) | Pearson | Spearman | RMSE   | MAE    |
> |----------------------|---------|----------|---------|----------|--------|--------|
> | MSM-Mut (w/o retrieval) | 0.4325  | 0.4031   | 0.6233  | 0.4954   | 1.6076 | 1.2155 |
>  | MSM-Mut (CBAE retrieval) | 0.4619   |  0.4262   |  0.6524   |  0.5158   |  1.5531   |  1.1622
>  MSM-Mut              | 0.4736  | 0.4354   | 0.6814  | 0.5786   | 1.4703 | 1.0212 |
>
> Task: S669
>
>   | Method               | Pearson | RMSE|
> |----------------------|---------|----------|
> | MSM-Mut (w/o retrieval) | 0.45  | 1.73   |
> | MSM-Mut (CBAE retrieval) | 0.52  | 1.55   |
> | MSM-Mut | 0.54  | 1.51   |
>
> Our experiments showed that structures retrieved using ESM-IF performed better than this simpler method. ESM-IF can capture non-adjacent backbone atom positions to some extent. However, the differences between these methods are limited because we observed that structures retrieved using ESM-IF embeddings also ensure a high degree of backbone similarity.
>
> **2.Expanding the database**
>
> Given the constraints of time, it is challenging to incorporate the entire set of predicted structures into our analysis. We are currently working on reducing the dimensionality of the embeddings through contrastive learning. However, the size of the entire AFDB dataset is substantial. To illustrate the impact of dataset size on model performance, we present results based on random selections of approximately one-tenth, one-hundredth, and one-thousandth of the database. Specifically, we randomly selected 16 motifs from the top 100, 1000, and 10,000 motifs, simulating a reduction of the database to approximately one-tenth, one-hundredth, and one-thousandth of its original size. Our observations indicate that within the existing high-quality structure database, the size of the dataset is indeed critical, as it significantly influences the availability of high-quality data.
>
> Task:S669
>
>   | Method               | Pearson | RMSE|
> |----------------------|---------|----------|
> | MSM-Mut (w/o retrieval) | 0.45  | 1.73   |
> | MSM-Mut (random in top 10000) | 0.43  | 2.03   |
> | MSM-Mut (random in top 1000) | 0.52  | 1.57   |
> | MSM-Mut  (random in top 100) | 0.53  | 1.59   |
> | MSM-Mut (top 16 neighbor)| 0.54  | 1.51   |
>
> **3. Adapting the Method for Different Types of Mutations**
>
> In the manuscript, we have already tested our method on several datasets, including SKEMPI2.0 (general surface mutation ΔΔG prediction), SARS-CoV-2 (antibody binding affinity prediction), S669 (mutation effects on stability), and novo-enzyme mutation effects on stability. These datasets cover a broad range of mutation effect prediction scenarios. If there are additional datasets or settings where you would like to see our method evaluated, we would be more than happy to provide those results.
>
> We have addressed the key weaknesses you pointed out in our paper. If these responses adequately resolve your concerns, we kindly ask you to consider slightly adjusting your evaluation. We are also open to further discussing any remaining issues you might have.

---

> > ### Comment · Reviewer_pY2f · 2024-08-14
> >
> > Thanks for the detailed rebuttal. I have raised the score.

---

### Official Review · Reviewer_4tC9 · 2024-07-14

**Soundness:** 4
**Presentation:** 2
**Contribution:** 2
**Rating:** 5
**Confidence:** 5

**Summary:**

- The paper presents a novel retrieval-augmented framework to efficiently retrieve similar local structure motifs in protein sequences for mutation effect prediction
- Current methods to understand coevolutionary patterns include MSA and domain-level structure clustering, which serve as a global representation of the protein and its evolutionary couplings
- The paper introduces Structure Motif Embedding Database (SMEDB), constructed from the ESM-IF structure-based embedding approach, to enable rapid GPU-accelerated kNN search and retrieve local structure motifs similar to those affected by mutations
- The retrieval method Multiple Structure Motif Alignment (MSMA) leverages embeddings that capture local coevolutionary patterns
Multi-Structure Motif Invariant Point Attention (MSM-IPA) model aggregates retrieved coevolutionary motif information to predict changes in binding free energy (G) on protein surfaces to assess protein-protein interactions

**Strengths:**

- The main strength of this paper is its novelty and originality of the idea. It is intuitive to look for information about the effects of mutations locally in the protein structure
- The search approach identifies similar local structure motifs from structurally unrelated proteins which is interesting independently
- The paper demonstrate MSM-IPA’s performance on the benchmark datasets compared to existing methods, which shows how the retrieval of information provides an advantage in predicting mutation effects on protein-protein interactions
- The paper shows the real-world utility of MSM-IPA by optimizing antibody engineering for SARS-CoV-2 and predicting enzyme thermostability for a novel enzyme

**Weaknesses:**

- The main weakness of the work is in the evaluations.
- It is difficult from Table 2 to conclude the competitive performance of MSM-Mut as other DL-based methods are performing better?
- In Table 2, I did not understand the rationale for making some numbers bold. If the lowest number is better, then why are all numbers bold?
- It would make reading the paper much easier if Tables were referred to in the paper.
- Several important mutation prediction baselines, such as DeepSeqeice and EVE, are missing from Table 1. These and others are outlined and now standardized in the ProteinGym benchmark paper.
- Without a large-scale evaluation, it would be hard to assess the utility of the method and fully evaluate the contributions
- The interpretability of MSM-IPA’s predictions might be obscured because it incorporates multiple structure motifs
- Figure 3 could benefit from some labels for the reader to identify T chain 576, 5KOV, H chain 103, and 7FAE

**Questions:**

1. Given that ESM-IF is trained on "global" protein structure and not local, why is it intuitive to use ESM-IF as a tool to embed local structures?
2. Would a stand-alone embedding module trained entirely on local structures perform better as an alternative to ESM-IF within the MSM-IPA framework?

**Limitations:**

As discussed earlier in the weakness section, the main limitation of the paper is evaluation which currently do not fully support the main claim.

---

> ### Author Rebuttal · Authors · 2024-08-07
>
> **Question 1: It is difficult from Table 2 to conclude the competitive performance of MSM-Mut as other DL-based methods are performing better?**
>
> We apologize for any difficulty in concluding the competitive performance of MSM-Mut from Table 2. Our primary intention with this table was to demonstrate two key points.
> Firstly, despite employing a relatively simple model architecture, incorporating the structure motifs retrieved led to a notable improvement in the model's accuracy for predicting mutations on SARS-CoV-2. Our method achieves performance comparable to state-of-the-art models and reaches top 20% levels for four of the five beneficial mutations.
>
> Secondly, we observed that most DL-based models, including our non-retrieval version, tend to downplay the significance of the RH103M mutation in enhancing binding. However, with the incorporation of retrieved information, the importance of this binding increased significantly, placing it in the top 20%. This observation supports our subsequent case study on the RH103M mutation, highlighting the crucial role of retrieved structure motifs in accurately predicting binding ΔΔG.
>
> **Question 2: In Table 2, I did not understand the rationale for making some numbers bold. If the lowest number is better, then why are all numbers bold?**
>
> We apologize for the excessive bolding in the table. Here, we chose to bold all mutations ranked in the top 100. Note that there are a total of 494 mutations, so we roughly selected the top 20% of mutations to bold. This is because, in practical protein engineering processes, the workflow typically involves using deep learning models to screen for potentially top-ranking mutations, which are then sent to the laboratory for binding affinity measurements. Therefore, in our setting, we consider the top 20% of mutations as the set that can be forwarded to downstream wet lab testing, and any favorable mutation ranking within this part can be successfully detected.
>
> **Question 3: It would make reading the paper much easier if Tables were referred to in the paper.**
>
> Thank you for your valuable feedback. We apologize for not providing enough references to the tables in the paper. We agree that referring to the tables within the text will enhance the readability of the paper. We will ensure that all tables are appropriately referenced in the revised manuscript.
>
> **Question 4: Several important mutation prediction baselines, such as DeepSeqeice and EVE, are missing from Table 1. These and others are outlined and now standardized in the ProteinGym benchmark paper.**
>
> Our tables in the paper already include the performance of most model types on the SKEMPI2.0 dataset. Since SKEMPI2.0 is a dataset for predicting the ΔΔG of surface mutations, alignment-based sequence methods like DeepSequence and EVE tend to perform relatively poorly. To provide a more comprehensive comparison, we have added the performance of EVE, ESM2, and supervised ESM2 in the table below. We will update this table in the revised manuscript.
> Task: SKEMPI2.0
> | Category       | Method               | Pearson (P.S.) | Spearman(P.S.) | Pearson | Spearman | RMSE   | MAE    |
> |----------------|----------------------|---------|----------|---------|----------|--------|--------|
> | Unsupervised   | ESM2               | 0.0100  | 0.0100   | 0.1700  | 0.1630   | 2.6580 | 2.0210 |
> |                | EVE          | 0.1131   |  0.0898   |  0.1237   |  0.1088   |  2.2622   |  1.4178   |
> |    Supervised  |     ESM2(Sup)      | 0.3330  | 0.3040   | 0.6030  | 0.5290   | 2.1500 | 1.6700 |
>
> **Question 5: Without a large-scale evaluation, it would be hard to assess the utility of the method and fully evaluate the contributions**
>
> In the field of ΔΔG prediction for surface mutations, the most commonly used dataset is SKEMPI2.0, which includes various types of protein-protein interface mutation data such as antibody-antigen (AB/AG), protease-inhibitor (Pr/PI), and T-cell receptor-major histocompatibility complex (TCR/pMHC).
> For the stability prediction, we followed the approach from the Stability Oracle[ref], which represents the current state-of-the-art for structure-based methods. This work used the cDNA dataset[ref] for pretraining and tested on the s669 dataset. We chose s669 as our stability benchmark because it was the most challenging dataset in the Stability Oracle study, with a Pearson correlation of only 0.52. Therefore, we aimed to optimize our method on this difficult dataset.
> We did not evaluate our method on the large-scale ProteinGym dataset because our method requires a pretraining dataset to enable MSM-IPA to integrate neighboring information. The cDNA pretraining dataset overlaps significantly with the ProteinGym test set. Removing the overlapping training data would substantially reduce the size of ProteinGym. Additionally, the data in ProteinGym are relatively simple, so we did not choose it for testing.
>
> **Question 6: The interpretability of MSM-IPA’s predictions might be obscured because it incorporates multiple structure motifs**
>
> We have provided the performance results using only one neighbor, which also shows significant improvement. This indicates that the most similar local structure motif we selected can serve as a potential post-mutation structure, contributing to the improved results.
> We aim to demonstrate through MSM-IPA experiments that our retrieval database contains such valuable information, which can be utilized beyond just MSM-IPA.
> Task: S669
>   | Method               | Pearson | RMSE|
> |----------------------|---------|----------|
> | MSM-Mut (w/o retrieval) | 0.45  | 1.73   |
> | MSM-Mut (1 neighbor) | 0.49  | 1.62   |
> | MSM-Mut (16 neighbor)| 0.54  | 1.51   |

---

> ### Author Response · Authors · 2024-08-07
> **More Rebuttal Information**
>
> **Question 7: Figure 3 could benefit from some labels for the reader to identify T chain 576, 5KOV, H chain 103, and 7FAE**
>
> In Figure 3, blue represents 5KOV and green represents 7FAE. The red segment is 5KOV's T chain 574-578, and the yellow segment is 7FAE's H chain 101-105.
>
> To help understand this, we extracted the local structure motifs (Rebuttal Fig. 1). Despite overall differences, their local backbone atoms and surrounding non-charged polar amino acids like serine and asparagine are very similar.
>
> **Question 8: Given that ESM-IF is trained on "global" protein structure and not local, why is it intuitive to use ESM-IF as a tool to embed local structures?**
>
> We chose ESM-IF for embedding because, although the embedding is calculated based on global structure, it is used for protein inverse-folding, which is sensitive to local structure. Preliminary case studies showed that ESM-IF embeddings help retrieve structures with similar nearby backbone frames. See Rebuttal Fig. 2 in the supplementary PDF.
>
> **Question 9: Would a stand-alone embedding module trained entirely on local structures perform better as an alternative to ESM-IF within the MSM-IPA framework?**
>
> Firstly, we aim to demonstrate that our method outperforms simple local backbone encoding. To compare, we tried a simpler method: for an amino acid, we defined its embedding as [$\phi_{i-4}, \psi_{i-4}, ..., \phi_{i+4}, \psi_{i+4}$], an 18-dimensional vector of angles for this amino acid and its four sequential neighbors.  We call this embedding Continuous Backbone Angle Embedding (CBAE). This method ensures similar sequential-local backbone structures. We tested performance on the SKEMPI and S669 datasets:
>
> Task: SKEMPI 2.0
>
> | Method| Pearson (P.S.) | Spearman(P.S.) | Pearson | Spearman | RMSE | MAE|
> |----|---|----|-----|-----|-----|---|
> | MSM-Mut (w/o retrieval) | 0.4325| 0.4031| 0.6233| 0.4954 | 1.6076 | 1.2155 |
>  | MSM-Mut (CBAE retrieval) | 0.4619|0.4262 |0.6524 |0.5158 |1.5531 |1.1622 |
>  |MSM-Mut | 0.4736| 0.4354 | 0.6814| 0.5786 | 1.4703 | 1.0212 |
>
> Task: S669
>
> | Method| Pearson | RMSE|
> |--|---|--|
> | MSM-Mut (w/o retrieval) | 0.45| 1.73 |
> | MSM-Mut (CBAE retrieval) | 0.52| 1.55 |
> | MSM-Mut | 0.54| 1.51 |
>
> Our experiments showed that ESM-IF performed better than this simpler method, capturing non-adjacent backbone atom positions. Training a good pre-trained local structure encoder is still an open question. Different tasks may benefit from task-specific embeddings. For our mutation effect prediction, we used embeddings from the commonly used ESM-IF model.

---

> > ### Comment · Reviewer_4tC9 · 2024-08-11
> > **Thanks for the notes**
> >
> > I thank the authors for the notes. My main criticism regarding choosing a global embedding (ESM-IF) to obtain local information is the difficulty of interpretability and lack of comparison with baseline stands. At this point, I cannot be more positive than a score of 5. I hope the comments help improve the paper.

---

> > > ### Author Response · Authors · 2024-08-13
> > >
> > > We sincerely appreciate the reviewer’s valuable suggestions. Regarding your concerns about interpretability and the lack of baseline comparisons, we would like to provide the following clarifications:
> > >
> > > **Interpretability**: First, we would like to explain why we chose ESM-IF, a global embedding, to model local information. Our motivation stems from the fact that the Protein Data Bank (PDB) is sufficiently large, leading us to believe that local structures of proteins are relatively dense. Thus, we aimed to directly construct a local-structure-based profile that could model the effects of mutations, particularly those on the protein surface, something that conventional MSA profiles cannot achieve. ESM-IF is trained on an inverse folding task, where the internal embeddings for each amino acid align well with our task. Therefore, we hypothesized that the similarity between these embeddings could, to some extent, quantify the similarity between local structures.
> > >
> > > Additionally, we have provided evidence of the validity of the information retrieved by our method. In Rebuttal Fig. 2, we display the retrieved structures, showing that the local motifs with the most similar embeddings indeed share highly similar backbone structures.
> > >
> > > **Baseline Comparisons**: We have already added several baseline experiments on the SKEMPI and S669 datasets, as mentioned earlier in the rebuttal. We also explained the reasons for not selecting the two large-scale datasets you mentioned. Currently, we are conducting tests of our method and baselines on a sub-dataset of ProteinGym that has no overlap with our fine-tuning training set. We will include these results in the appendix of the manuscript.
> > >
> > > We hope that these clarifications address your concerns. If you have any further questions, we would be more than happy to continue the discussion with you.

---

### Author Rebuttal · Authors · 2024-08-07

Thank you for your insightful and constructive comments as well as your appreciation of our work. Below are some clarifications and answers to your questions. If our response does not fully address your concerns, please post additional questions and we will be happy to have further discussions. As we are not able to update the new draft during the rebuttal stage, we promise to update our draft based on your suggestions if accepted.

---

### Decision · Program_Chairs · 2024-09-25

**Decision:**

Accept (poster)

**Comment:**

This paper presents a retrieval-augmented framework for protein mutation prediction. Structural motifs are first retrieved according to the structural similarity with the query structural, measured by a protein structure encoder. Experimental results on multiple protein mutation prediction tasks prove the effectiveness of the proposed approach.

Overall, leveraging structural similar motif for mutation prediction is a very interesting and novel idea. Increasing the interpretability of the proposed approach and also validate the model on more benchmarks (e.g., ProGym) will further strengthen the paper.